# Molecular profiling of the artemisinin resistance Kelch 13 gene in *Plasmodium falciparum* from Nigeria

Fehintola V. Ajogbasile[1,2], Paul E. Oluniyi[1,2], Adeyemi T. Kayode[1,2], Kazeem O. Akano[1,2], Benjamin B. Adegboyega[1], Courage Philip[1], Nnenna Ogbulafor[3], Henrietta U. Okafor[4], Stephen Oguche[5], Robinson D. Wammanda[6], Olugbenga A. Mokuolu[7], Onikepe A. Folarin[1,2], Christian T. Happi[1,2]*

1 African Center of Excellence for Genomics of Infectious Diseases (ACEGID), Redeemer's University, Ede, Osun State, Nigeria, 2 Department of Biological Sciences, Faculty of Natural Sciences, Redeemer's University, Ede, Osun State, Nigeria, 3 National Malaria Elimination Programme, The Federal Ministry of Health, Abuja, Nigeria, 4 Department of Pediatrics, Institute of Child Health, University of Nigeria Teaching Hospital, Enugu, Nigeria, 5 Department of Paediatrics, Faculty of Clinical Sciences, College of Health Sciences, University of Jos/Jos University Teaching Hospital, Jos, Nigeria, 6 Department of Paediatrics, Ahmadu Bello University, Zaria, Nigeria, 7 Department of Paediatrics and Child Health, University of Ilorin, Ilorin, Nigeria

* happic@run.edu.ng

**Data Availability Statement:** All sequences from this study are available on GenBank with accession numbers MT113570 - MT113901.

## Abstract

Accurate assessment and monitoring of the *Plasmodium falciparum* Kelch 13 (*pfk13*) gene associated with artemisinin resistance is critical to understand the emergence and spread of drug-resistant parasites in malaria-endemic regions. In this study, we evaluated the genomic profile of the *pfk13* gene associated with artemisinin resistance in *P. falciparum* in Nigerian children by targeted sequencing of the *pfk13* gene. Genomic DNA was extracted from 332 dried blood (DBS) spot filter paper samples from three Nigerian States. The *pfk13* gene was amplified by nested polymerase chain reaction (PCR), and amplicons were sequenced to detect known and novel polymorphisms across the gene. Consensus sequences of samples were mapped to the reference gene sequence obtained from the National Center for Biotechnology Information (NCBI). Out of the 13 single nucleotide polymorphisms (SNPs) detected in the *pfk13* gene, five (F451L, N664I, V487E, V692G and Q661H) have not been reported in other endemic countries to the best of our knowledge. Three of these SNPs (V692G, N664I and Q661H) and a non-novel SNP, C469C, were consistent with late parasitological failure (LPF) in two States (Enugu and Plateau States). There was no validated mutation associated with artemisinin resistance in this study. However, a correlation of our study with *in vivo* and *in vitro* phenotypes is needed to establish the functional role of detected mutations as markers of artemisinin resistance in Nigeria. This baseline information will be essential in tracking and monitoring *P. falciparum* resistance to artemisinin in Nigeria.

**Funding:** This work is made possible by support from Flu Lab and a cohort of generous donors through TED's Audacious Project, including the ELMA Foundation, MacKenzie Scott, the Skoll Foundation, and Open Philanthropy. This work was supported by grants from the National Institute of Allergy and Infectious Diseases (https://www.niaid.nih.gov), NIH-H3Africa (https://h3africa.org) (U01HG007480 and U54HG007480 to C.T.H), the World Bank grant (worldbank.org) (ACE IMPACT project) to C.T.H. The U.S President's Malaria Initiative (USPMI) funded the primary drug efficacy study from which samples were obtained for the current study. The funders had no role in study design, data collection and analysis, decision to publish, or preparation of the manuscript.

**Competing interests:** The authors have declared that no competing interests exist.

## Introduction

Malaria, one of the major global health challenges, has co-existed with humans for over 40 centuries. Control of this ancient disease heavily relies upon the use of antimalarial drugs [1]. Artemisinin-based combination therapies (ACTs) [e.g., artemether-lumefantrine (AL) and artesunate-amodiaquine (AA)] are the current line of treatment for malaria. These drugs were recommended by the World Health Organization (WHO) as first-line treatment for uncomplicated falciparum malaria in 2001 and have since been widely adopted to treat falciparum malaria globally [2]. Artemisinin (ART)-resistant *P. falciparum* has been confirmed to have emerged from the Greater Mekong Subregion (GMS) Thai-Cambodian border [3, 4] and has spread to other malaria endemic regions [5–8].

Several point mutations in the *pfk13* gene (F446I, N458Y, M476I, Y493H, R539T, I543T, P553L, R561H, P574L and C580Y) have been validated to correlate with clinical ART resistance in Southeast Asia and South America [2, 9] and confirmed to confer elevated survival rates based on Ring-stage Survival Assays (RSA)0–3 h [10–12]. Furthermore, mutations and increased copy numbers of genes such as *P. falciparum* multidrug resistant gene-1 (*pfmdr1*) and *P. falciparum* chloroquine resistance transporter (*pfcrt*) gene have been linked to resistance in artemisinin (and derivatives) partner drugs such as lumefantrine, amodiaquine, mefloquine and piperaquine [13–18]. Due to these mutations, the efficacy of ACTs may be compromised [19]. Recently for the first time in Africa, two of the validated mutations (R561H and P574L) on the *pfk13* gene have been reported in Rwanda to be associated with *in vitro* resistance to ACTs [7, 8]. However, in other sub-Saharan African countries such as Nigeria, no association between ART-resistant parasites and the ten validated mutations has been found [20–23].

Despite the need to regularly survey for the emergence of these *pfk13* mutant alleles in different malaria-endemic regions of Nigeria, only a small number of systematic molecular epidemiological studies on field *P. falciparum* isolates have so far been conducted [2]. A few studies have reported cases with delayed response to ACTs [22, 23], and a sporadic scan for amino acid mutations in the *pfk13* gene identified three nonsynonymous mutations (G592R, Q613H, and G665S) and other synonymous mutations [9].

This study aims to describe the potential emergence and spread of ART-resistance alleles in the *pfk13* gene in three Nigerian States. The study was designed to inform malaria policy-makers and research scientists in accordance with the objectives of the therapeutic efficacy study (TES) of the National Malaria Elimination Program (NMEP) of the Federal Republic of Nigeria recommended by the WHO.

## Materials and methods

### Study site

This is a retrospective, cross-sectional, community-based study which is part of the TES for monitoring antimalarial efficacies of Artesunate-amodiaquine (AA), Artemether-lumefantrine (AL) and Dihydroartemisinin-piperaquine (DHP) in the treatment of uncomplicated *P. falciparum* infections in children aged 6–96 months old in Nigeria. A cohort of 586 children was enrolled from three sentinel sites of the 2018 antimalarial drug efficacy testing and monitoring of the NMEP of the Federal Ministry of Health, namely; Kura (n = 200), Barkin Ladi (n = 185) and Agbani (n = 201), in Kano, Plateau and Enugu States respectively. Full description of study sites is available online at https://www.health.gov.ng/doc/2018-TES-FINAL-REPORT.pdf

## Enrolment and sample collection

Children were enrolled if they were 6–96 months old. Children were eligible for enrolment if they had symptoms compatible with acute uncomplicated malaria, *P. falciparum* mono-infection with parasite count ranging from 2,000–200,000 asexual forms/μl by microscopy and body (axillary) temperature of $\geq$ 37.5 ˚C or a history of fever in the 24 hours preceding presentation. Children with severe malaria, severe malnutrition, serious underlying diseases (renal, cardiac, or hepatic diseases), and known allergies to the study drugs were excluded. Follow-up clinical and parasitological evaluations were done daily on days 1 to 3 and then on days 7, 14, 21, 28, 35, and 42.

Two to three drops of finger-pricked blood samples were blotted on 3mm Whatman filter paper (Whatman International Limited, Maidstone, United Kingdom) before treatment initiation (Day 0) and post-treatment initiation on days 7, 14, 21, 28, 35 and 42. Blood samples impregnated on filter papers were allowed to air-dry appropriately at room temperature and kept in airtight envelopes with silica gel at room temperature until analysed. A total of 586 DBS filter paper samples from children were sent to the African Centre of Excellence for Genomics of Infectious Diseases (ACEGID), Redeemer's University, for molecular analysis. Samples were collected during the intense malaria season (August-October) of 2018.

## Ethical declaration

The study was conducted in accordance with the Declaration of Helsinki, and the protocol was approved by the National Health Research Ethics Committee, Federal Ministry of Health (FMOH), Abuja, Nigeria. Written informed consent was obtained from parents/guardians for children prior to enrollment in this study. Child assent was obtained from children aged 84–96 months.

## Assessment of treatment outcome

Response to drug treatment was evaluated using the following treatment indices:

1. Adequate clinical and parasitological response (ACPR), early treatment failure (ETF), late clinical failure (LCF) and late parasitological failure (LPF) [Full description of WHO TES study protocol available at https://www.who.int/docs/default-source/documents/publications/gmp/methods-for-surveillance-of-antimalarial-drug-efficacy.pdf?sfvrsn=29076702_2].

2. Asexual parasite reduction ratio (PRR) one or two days after treatment initiation (PRRD1 or PRRD2) defined as the ratio of asexual parasitaemia pre-treatment initiation and that on day one or two, respectively.

3. Asexual parasite positivity on day 1 (APPD1), 2 (APPD2) or 3 (APPD3) is defined as the proportion of children with residual asexual parasitaemia one, two or three days after treatment initiation, respectively Asexual parasite clearance time (PCT) defined as the time elapsing between drug administration and absence of microscopic detection of viable asexual parasitaemia.

## Parasite genomic DNA extraction

A total of 300 pre-treatment (Day 0) DBS filter paper samples from all three sites were utilised in this study. Of the 300 samples, 32 had LPF making a final total of 332 samples selected for

analysis. Parasite genomic DNA was extracted from 332 DBS samples using the Zymo Quick-DNA Miniprep Plus kit (17062 Murphy Avenue Irvine, California 192614, United States of America) according to the manufacturer's protocol.

## Amplification of *pfk*13 gene and targeted sequencing

The *pfk13* propeller domain was amplified by nested PCR using PuReTaq Ready-To-Go PCR beads (GE Healthcare UK Limited) according to the manufacturer's protocol. We used primers *kelch*-outer-F 5′-gggaatctggtggtaacagc-3′ and *kelch*-outer-R 5′-cggagtgaccaaatctggga- 3′ for the primary PCR, and *kelch*-inner-F 5′-gccttgttgaaagaagcaga-3′ and *kelch*-inner-R 5′-gccaagctgccattcatttg-3′ for the nested PCR. The nested PCR product was 849 bp and corresponds to nucleotide sequence 1279–2127 (representing codons 427–709) of PF3D7_1343700 *K13* propeller domain, which included mutations correlated with delayed parasite clearance [10].

The ready-to-go PCR beads were reconstituted to a final volume of 20 μL in the primary reaction, 2 μL of DNA was amplified with 0.5 μM of each primer. Cycling conditions were 95 ˚C for 1 minute, followed by 35 cycles at 95˚C for 20 seconds, 58 ˚C for 20 seconds, and 60 ˚C for 1 minute, with a final extension at 60 ˚C for 3 minutes.

One microlitre of the primary reaction was further amplified with 0.5 μM of each primer in the nested PCR reaction. Cycling conditions were 95 ˚C for 1 minute, followed by 35 cycles at 95 ˚C for 20 seconds, 56 ˚C for 20 seconds, and 60 ˚C for 1 minute, with a final extension at 60 ˚C for 3 minutes. Nested amplicons were analysed by electrophoresis on a 2% agarose gel to confirm amplification. We purified the nested amplicons using ExoSAP-IT (Affymetrix, Santa Clara, CA, USA), and we sequenced them using BigDye Terminator v.1.1 (Life Technologies, Carlsbad, CA, USA) and the same primers as for nested PCR (kelch-inner-F and kelch-inner-R). We carried out sequencing using an Applied Biosystems 3500 XL series Genetic Analyser at ACEGID, Redeemer's University, Ede, Osun State, Nigeria.

## Data analysis

**Identification of polymorphisms.** Chromatograms of the individual sequences were viewed using Geneious v2020.0.4 [24, 25] and manual base calling was carried out as needed for some of the sequences. Consensus sequences were generated for all samples. The consensus sequences were generated by first aligning the forward and reverse reads for individual samples using Geneious alignment. A consensus was reached from the resulting alignment by choosing the highest quality base and also carrying out manual base calling for regions of ambiguities. To detect polymorphisms in the sequences encoding the *pfk13* propeller gene, we obtained the reference nucleotide sequence of the *pfk13* gene from the NCBI database (PF3D7_1343700 sequence region spanning region 1,724,817–1,726,997 bp of chromosome 13). We mapped the consensus sequences of the samples to the reference gene sequence using Geneious v2020.0.4.

**Statistical analysis of treatment outcome.** Discrete variables (such as proportions of frequencies) were compared by calculating $\chi2$ using Yates' correction, Fisher's exact or Mantel Haenszel tests. Normally distributed, continuous data were analysed by Student's t-test or analysis of variance (ANOVA) as it is applicable. Mann–Whitney U tests or Kruskal Wallis tests (or by Wilcoxon ranked sum test) was used to compare data that did not conform to normal distribution. P values of <0.05 were taken to indicate significant differences.

**Table 1. Demographic characteristics of children at enrollment.**

| Parameter | Enugu | Kano | Plateau | All | P value |
|---|---|---|---|---|---|
| n | 100 | 100 | 100 | 300 | |
| **Gender** | | | | | |
| M: F | 55:45 | 60:40 | 50:50 | 165:135 | 0.36 |
| **Age (Months)** | | | | | |
| Mean | 57.69 | 51.7 | 66.54 | 58.62 | <0.0001 |
| 95% CI | 52.33–63.05 | 46.6–56.8 | 62.24–70.85 | 55.71–61.52 | |
| ≤ 60months | 60 | 70 | 47 | 177 | 0.004 |
| **Asexual Parasitemia** | | | | | |
| Geometric mean | 17098 | 14526 | 22984 | 17765 | 0.046 |
| 95% CI | 13060–22384 | 12130–17395 | 17801–29676 | 15498–20363 | |
| ≥ 100,000 | 10 | 3 | 16 | 29 | 0.004 |

## Results

### Demographics

Of the 300 participants' samples analysed, 165 (55%) were male. The mean age of all children was 58.42±25.71 months (95% confidence interval (CI) 55.71–61.52). The geometric mean of the asexual parasitemia was 17,765 $\mu L^{-1}$ (95%CI 15498–20363) (Table 1). Children enrolled in Kano State were significantly (p<0.0001) younger and had significantly (p = 0.046) lower enrollment asexual parasitaemia compared to children enrolled in other States (Table 1).

### Study treatment outcome

The overall ACPR_c values for AA, AL and DHP in Enugu, Kano and Plateau States were 100%, 99.3% and 100% respectively. The ACPR_c values for AA in Kano and Plateau States were 100% respectively while the ACPR_c values for AL in Enugu, Kano and Plateau States were 98%, 100% and 100% respectively. Only Enugu State tested DHP and recorded an ACPR_c value of 100%. Of the 32 presented with LPF, 14 occurred in Enugu, 11 in Kano and 8 in Plateau State (Table 2).

### *Pfk13* gene mutations

We amplified and sequenced the *pfk13* gene from all the 332 samples (300 pre-treatment samples and 32 LPF samples). Thirteen *pfk13* gene mutations were detected in 21 out of the 332 sequences analysed in this study (Table 3). The prevalence of parasites with mutations on chromosome 13 in the propeller region of the *pfk13* protein was 6.3%, 93.7% did not have mutations (Fig 1). The highest occurring mutation Q613H (1.5%) was detected in five samples (three from Plateau State and two from Enugu State). The C469C mutation with 1.2% was detected in four samples from Enugu State. Mutations V692G and G449S were detected in 0.6% of samples (G449S in Kano State; V692G in Enugu and Plateau States). We summarized mutations detected in this study and previous Nigerian studies in Table 4.

### Treatment outcome of children infected with mutated *P. falciparum*

Characteristics of the children with *P. falciparum* with *pfk13* mutations are shown in Table 5. Children infected with the mutated parasites are relatively older children (mean age: 53.7 ± 22 months, 14 of 21 were aged ≥48 months). They were also characterized by low enrollment

**Table 2. Summary of treatment outcome by State and drug.**

| State | Treatment Outcome | Drug | | | Total |
|---|---|---|---|---|---|
| | | AA | AL | DHP | |
| Enugu | Number of samples | | 50 | 50 | 100 |
| | ETF | | 0 | 0 | 0 |
| | LCF | | 0 | 0 | 0 |
| | LPF | | 11 | 2 | 14 |
| | ACPR_u | | 39 | 48 | 87 |
| | ACPR_c | | 49 | 50 | 99 |
| | %ACPR_c | | 98 | 100 | 99 |
| Kano | Number of samples | 50 | 50 | | 100 |
| | ETF | 0 | 0 | | 0 |
| | LCF | 0 | 0 | | 0 |
| | LPF | 2 | 9 | | 11 |
| | ACPR_u | 48 | 41 | | 89 |
| | ACPR_c | 50 | 50 | | 100 |
| | %ACPR_c | 100 | 100 | | 100 |
| Plateau | Number of samples | 50 | 50 | | 100 |
| | ETF | 0 | 0 | | 0 |
| | LCF | 0 | 0 | | 0 |
| | LPF | 1 | 7 | | 8 |
| | ACPR_u | 49 | 43 | | 92 |
| | ACPR_c | 50 | 50 | | 100 |
| | %ACPR_c | 100 | 100 | | 100 |
| Total | Number of samples | 100 | 150 | 50 | 300 |
| | ETF | 0 | 0 | 0 | 0 |
| | LCF | 0 | 0 | 0 | 0 |
| | LPF | 3 | 27 | 2 | 32 |
| | ACPR_u | 97 | 123 | 47 | 267 |
| | ACPR_c | 100 | 149 | 50 | 299 |
| | %ACPR_c | 100 | 99.3 | 100 | 99.7 |

Crude ACPR (ACPR_u); PCR-corrected ACPR (ACPR_c).

WHO protocol for parasite genotyping to differentiate recrudescence from new infections available at http://whqlibdoc.who.int/publications/2008/9789241596305_eng.pdf.

asexual parasitaemia (geometric mean parasitaemia: 17,530 $\mu L^{-1}$; 16 of 21 children had enrollment asexual parasitaemia $< 50,000 \mu L^{-1}$), relatively slow clearance time (two-third of the cohort cleared parasitaemia by after day 1). Following artemisinin-based combination treatment of the uncomplicated infection, 7 of 21 children (33%) had recurrent parasitaemia within 21–42 days of follow-up post treatment initiation (mean time to recurrence: 29±6.3 days), of these seven, only one child from Enugu State had recurrent parasitaemia due to recrudescence while four children from Enugu State and two from Plateau State had recurrent parasitaemia due to reinfection. It appears LPF was consistent with C469C (predominant in Enugu State) and V692G (observed in Enugu and Plateau States). Also, the Q613H mutation occurring in these States had ACPR phenotype.

**Table 3. *Pfk13* polymorphisms observed in Enugu, Kano and Plateau States.**

| S/N | K13 Amino Acid Locus | Nucleotide Locus | Reference Allele | Mutant Allele | Mutation Type | Country Previously Observed | Nigerian State Observed |
|---|---|---|---|---|---|---|---|
| 1. | K438N | 1314 | A | T | Non-synonymous | SE Asian Countries | Kano |
| 2. | G449S | 1345 | G | A | Non-synonymous | Mali | Kano |
| 3. | F451L | 1353 | T | - | Non-synonymous | - | Kano |
| 4. | C469C | 1407 | C | T | Synonymous | Kenya, Malawi, Senegal, Niger, Congo, DRC | Enugu |
| 5. | V487E | 1460 | T | A | Non-synonymous | - | Enugu |
| 6. | A557S | 1669 | G | T | Non-synonymous | Congo, DRC, Côte d'Ivoire | Enugu |
| 7. | A578S | 1732 | G | T | Non-synonymous | Uganda, Kenya, DRC, Gabon, Mali, Ghana, Cameroon, Cambodia, India | Enugu |
| 8. | Q613H | 1839 | A | T | Non-synonymous | Senegal, Ghana, Tanzania | Enugu & Plateau |
| 9. | A621A | 1863 | T | A | Synonymous | - | Enugu |
| 10. | Q661H | 1983 | A | - | Non-synonymous | - | Enugu |
| 11. | N664I | 1991 | A | T | Non-synonymous | - | Plateau |
| 12. | V692G | 2075 | T | G | Non-synonymous | - | Enugu & Plateau |
| 13. | N694K | 2082 | T | A | Non-synonymous | Angola, Cote d'Ivoire | Kano |

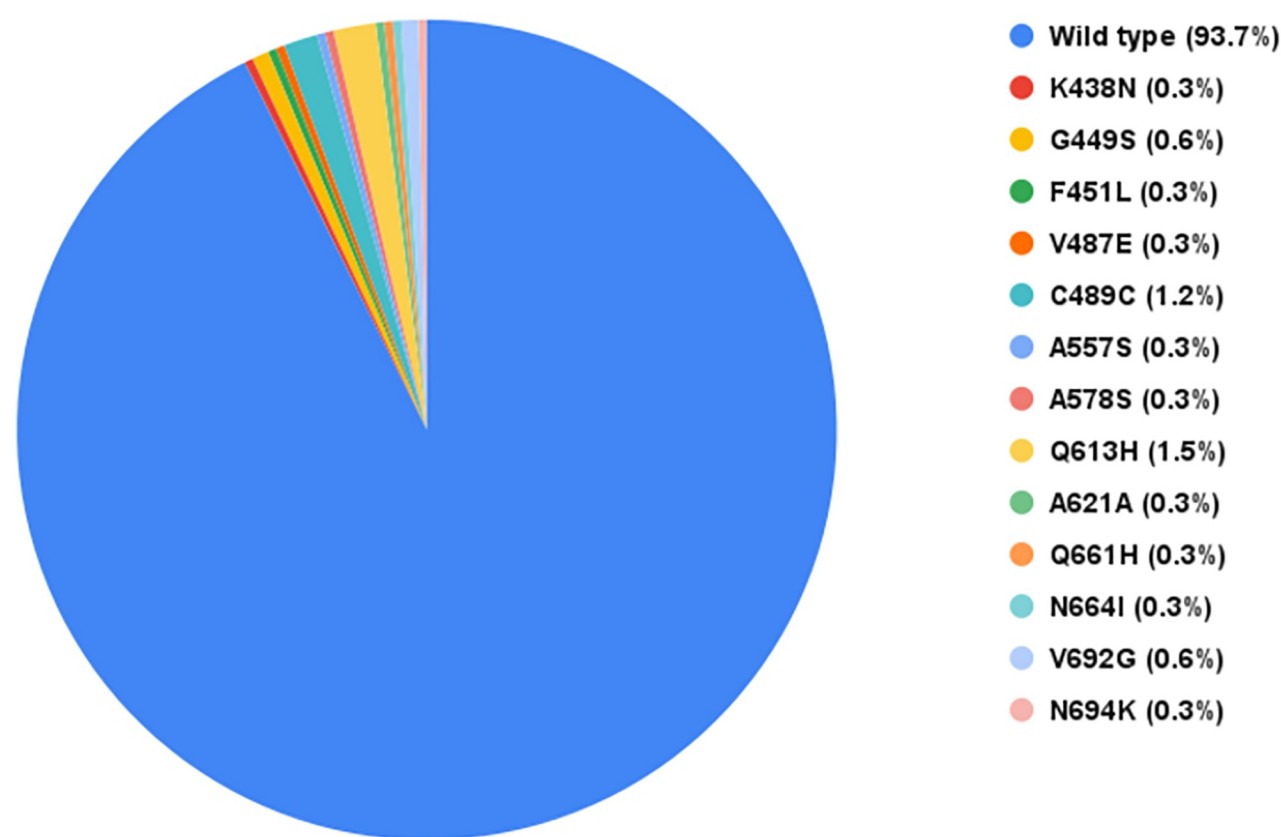

**Fig 1. A chart showing the frequency distribution of *pfk13* gene polymorphisms observed in this study.**

**Table 4. Molecular surveillance of *Pfk13* propeller polymorphisms in Nigeria.**

| S/N | Mutations Previously observed | Year observed | Reference |
|-----|-------------------------------|---------------|-----------|
| 1 | H136N | 2015, 2016 | [26] |
| 2 | K189T | 2015, 2016 | [26] |
| 3 | E433G | 2018 | [27] |
| 4 | F434I | 2018 | [27] |
| 5 | F434S | 2018 | [27] |
| 6 | K438N | 2018 | Observed in this study |
| 7 | P441S | 2017, 2018 | [28] |
| 8 | F442F | 2018 | [27] |
| 9 | G449S | 2018 | Observed in this study |
| 10 | F451L | 2018 | Observed in this study |
| 11 | D464N | 2010, 2011 | [29] |
| 12 | C469C* | 2017 | [28] |
| 13 | V487E | 2018 | Observed in this study |
| 14 | F492F | 2018 | [27] |
| 15 | G496G | 2014 | [30] |
| 16 | V510V | 2016 | [20] |
| 17 | P553P | 2016 | [20] |
| 18 | A557S | 2018 | Observed in this study |
| 19 | A578S* | 2010, 2011 | [29] |
| 20 | V589V | 2016 | [28] |
| 21 | K610R | 2014 | [30] |
| 22 | Q613H* | 2010, 2011, 2015, 2016 | [29, 26] |
| 23 | A621A* | 2014 | [31] |
| 24 | A626T | 2014 | [30] |
| 25 | A627A | 2014 | [30] |
| 26 | V650F | 2016 | [28] |
| 27 | Q661H | 2018 | Observed in this study |
| 28 | N664I | 2018 | Observed in this study |
| 29 | N664N | 2017 | [28] |
| 30 | G665C | 2016 | [20] |
| 31 | V666V | 2016 | [20] |
| 32 | A676A | 2016 | [28] |
| 33 | I684N | 2018 | [27] |
| 34 | I684T | 2018 | [27] |
| 35 | E688K | 2018 | [27] |
| 36 | V692G | 2018 | Observed in this study |
| 37 | N694K | 2018 | Observed in this study |

*Mutations observed in previous studies and our study.

## Comparison of responsiveness indices following treatment initiation in children with and without mutated *Pfk13*

As a result of the small number of samples with mutations in our study (n = 21), samples with mutation were matched for age, gender, enrolment asexual parasitaemia, same day presentation and same treatment with those without. Following treatment intiation, treatment indices such as APPD1-2 and PRRD1-2 were similar in the 2 groups (P>0.2). None of the children

**Table 5. Clinical features of children with mutated *P falciparum* in this study.**

| Study site | Sample ID | Gender | Age (Month) | Enrollment asexual parasitaemia (uL$^{-1}$) | Mutation Position | Parasite clearance time (day) | Antimalarial Treatment | Treatment outcome |
|---|---|---|---|---|---|---|---|---|
| Enugu | 33 | M | 48 | 30571 | A578S | 1 | AL | ACPR |
| | 89 | F | 60 | 12817 | Q613H | 2 | AL | ACPR |
| | 106 | F | 36 | 16290 | Q613H | 2 | AL | ACPR |
| | 134 | M | 72 | 57850 | A621A | 2 | AL | ACPR |
| | 154 | M | 60 | 3424 | V487E | 2 | AL | ACPR |
| | 179 | M | 36 | 61320 | A557S | 2 | DHP | ACPR |
| | 197 | M | 48 | 2055 | C469C | 2 | DHP | LPF (D42) |
| | 228[+] | F | 24 | 91680 | C469C | 2 | AL | LPF (D21) |
| | 236 | M | 36 | 9794 | C469C | 2 | AL | LPF (D28) |
| | 239 | M | 72 | 6714 | V692G | 1 | AL | LPF (D28) |
| | | | | | Q661H | | | |
| | 254 | M | 96 | 29229 | C469C | 2 | AL | LPF (D28) |
| Kano | | | | | | | | |
| | 14 | M | 12 | 15923 | N694K | 2 | AA | ACPR |
| | 26 | F | 24 | 37880 | F451L | 1 | AL | ACPR |
| | 51 | F | 84 | 26523 | K438N | 2 | AA | ACPR |
| | 59 | M | 36 | 16888 | G449S | 2 | AL | ACPR |
| | 77 | M | 48 | 2250 | G449S | 1 | AA | ACPR |
| | | | | | | | | |
| Plateau | 31 | F | 84 | 26069 | Q613H | 1 | AA | ACPR |
| | 58 | M | 60 | 16304 | Q613H | 1 | AL | ACPR |
| | 62 | F | 72 | 61080 | Q613H | 2 | AL | ACPR |
| | 77 | F | 60 | 5428 | N664I | 1 | AL | LPF (D28) |
| | 155 | M | 60 | 54739 | V692G | 2 | AA | LPF (D28) |
| Mean (SD) | | | 53.71 (22) | 17, 530* | | 1.67 (0.48) | | 29 (6.3)[#] |

*Geometric mean;

[#]Mean LPF;

[+]Only sample with recrudescence recurrent parasitaemia.

with mutant parasites had persistent asexual parasitaemia two or three days after treatment initiation. In comparison, 28.6% (6 of 21) of children and 4.8% (1 of 21) of children of the cohort infected with non-mutant parasites had persistent asexual parasitaemia two and three days after treatment initiation respectively. PCT was significantly longer in children infected with non-mutant parasites than those infected with mutant parasites (2.3±1.2 days versus 1.7±0.5 days, respectively; P = 0.03).

## Discussion

The knowledge of mutations in the *pfk13* gene associated with slow clearance of artemisinin derivatives provides the ability to track the emergence and prevent the spread of resistant parasites and assess the effectiveness of control measures.

In this study, we detected a total of 13 SNPs (out of which 11 were non-synonymous) in the *pfk13* gene of *P falciparum* obtained from Nigerian children. None of them was among the ten SNPs (F446I, N458Y, M476I, Y493H, R539T, I543T, P553L, R561H, P574L and C580Y) which have been validated to be associated with artemisinin resistance in Southeast Asia, South

America and Rwanda [2, 6–9]. The absence of the validated mutations in our study is in line with many recent studies carried out in Africa [32–40] and in Nigeria [20, 27, 29, 34, 41, 42].

In addition, eight (A578S, C469C, Q613H, K438N, A621A, N694K, G499S and A557S) of the 13 SNPs observed in our study have also been observed in Southeast Asia and other sub-Saharan African countries [10, 28, 30, 31, 33, 34, 36, 39, 40, 43–47] as represented in Table 4. The Q613H was the most prominent SNP observed in our study and has been reported in recent studies in Nigeria [26, 29]. However, A578S is the most prominent SNP observed among sub-Saharan African countries like Ghana, Kenya, Gabon, DRC, Uganda, Cameroon and Mali [34, 39, 46]. This mutation (A578S) represents a change from a non-polar to a polar amino acid and can alter the shape of the *pfk13* protein in the regions it has been observed in [35]. Therefore, more attention must be paid to this mutation as it is emerging to be the most prominent mutation observed in the *pfk13* gene in sub-Saharan Africa [34, 39, 36].

We detected five novel SNPs (F451L, N664I, V487E, V692G and Q661H), which to the best of our knowledge, have not yet been described elsewhere. Three of these SNPs (V692G, N664I and Q661H) and a non-novel SNP, C469C were associated with LPF in two States (Enugu and Plateau States) (Table 5). More studies would have to be carried out to understand these mutations better and unravel their effects or importance.

Data reported in this study suggests that all the thirteen (13) mutations observed in our study are less likely to be associated with a delayed parasite clearance phenotype as evidenced by similar parasite reduction ratios and proportion with persistent asexual parasitaemia following treatment initiation. In addition, significantly longer asexual PCT in children infected with non-mutant *pfk13* parasites indicate that the mutants identified in the parasites circulating in Nigeria do not confer resistance to artemisinin derivatives.

As some of these mutations seem to be indigenous to African parasites, it is imperative that *in vivo* and *in vitro* studies are conducted to validate their possible roles in the emergence and spread of ART reduced susceptibility/resistance in Nigeria and sub-Saharan Africa as a whole as observed in previous studies [48, 49].

Although these polymorphisms reported in this study have not been associated with ART resistance, recent studies have shown that two of the validated mutations (R561H and P574L) associated with ART-resistance have been observed in Rwanda, a country which like Nigeria is in the Sub-saharan African Region [7, 8]. This is worrisome because it is only a matter of time before such mutant malaria parasites are introduced into Nigeria due to migration patterns between these two countries. With the emergence of these mutations in the *pfk13* gene, there is a need for constant and routine monitoring to avoid bad surprises and to have in place control strategies should resistant parasites emerge.

## Conclusion

There was no validated mutation associated with ART resistance in this study. However, we observed novel and other established mutations reported to be circulating in Nigeria and other African countries. Correlation of mutation data obtained in this study with *in vivo* and *in vitro* phenotypes is needed to establish the functional role of detected mutations as markers of artemisinin resistance in Nigeria.

## Acknowledgments

The authors thank all the patients, their parents or guardians for volunteering to participate in the study. We acknowledge technical support from colleagues at the ACEGID. We also acknowledge the principal investigators (PIs) in each of the three sentinel locations considered

in this study and the National Malaria Elimination Program of the Federal Ministry of Health in Nigeria.

## Author Contributions

**Conceptualization:** Fehintola V. Ajogbasile, Christian T. Happi.

**Data curation:** Fehintola V. Ajogbasile, Paul E. Oluniyi.

**Formal analysis:** Paul E. Oluniyi, Kazeem O. Akano.

**Funding acquisition:** Christian T. Happi.

**Investigation:** Fehintola V. Ajogbasile.

**Methodology:** Fehintola V. Ajogbasile, Paul E. Oluniyi.

**Project administration:** Onikepe A. Folarin, Christian T. Happi.

**Resources:** Nnenna Ogbulafor, Henrietta U. Okafor, Stephen Oguche, Robinson D. Wammanda, Olugbenga A. Mokuolu, Christian T. Happi.

**Software:** Paul E. Oluniyi.

**Supervision:** Onikepe A. Folarin, Christian T. Happi.

**Validation:** Fehintola V. Ajogbasile, Paul E. Oluniyi, Adeyemi T. Kayode, Kazeem O. Akano, Benjamin B. Adegboyega, Courage Philip.

**Visualization:** Fehintola V. Ajogbasile, Paul E. Oluniyi.

**Writing – original draft:** Fehintola V. Ajogbasile, Paul E. Oluniyi.

**Writing – review & editing:** Fehintola V. Ajogbasile, Paul E. Oluniyi, Adeyemi T. Kayode, Kazeem O. Akano, Benjamin B. Adegboyega, Courage Philip, Onikepe A. Folarin, Christian T. Happi.

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
