## [Decision Letter · Decision Letter 0]

11 Oct 2021

PONE-D-21-25383Molecular profiling of artemisinin resistance Kelch 13 gene in Plasmodium falciparum from NigeriaPLOS ONE

Dear Dr. Happi,

Thank you for submitting your manuscript to PLOS ONE. After careful consideration, we feel that it has merit but does not fully meet PLOS ONE’s publication criteria as it currently stands. Therefore, we invite you to submit a revised version of the manuscript that addresses the points raised during the review process.

We look forward to receiving your revised manuscript.

Kind regards,

Jutta Marfurt, PhD

Academic Editor

PLOS ONE

Journal Requirements:

a) Did participants provide their written or verbal informed consent to participate in this study?

This work is made possible by support from Flu Lab and a cohort of generous donors through TED’s Audacious Project, including the ELMA Foundation, MacKenzie Scott, the Skoll Foundation, and Open Philanthropy. This work was supported by grants from the National Institute of Allergy and Infectious Diseases (https://www.niaid.nih.gov), NIH-H3Africa (https://h3africa.org) (U01HG007480 and U54HG007480 to C.T.H), the World Bank grant (worldbank.org) (ACE IMPACT project) to C.T.H.The U.S President’s Malaria Initiative (USPMI) funded the primary drug efficacy study from which samples were obtained for the current study.

5. Thank you for stating the following in the Acknowledgments/ Funding Section of your manuscript: 

This work is made possible by support from Flu Lab and a cohort of generous donors through TED’s Audacious Project, including the ELMA Foundation, MacKenzie Scott, the Skoll Foundation, and Open Philanthropy. This work was supported by grants from the National Institute of Allergy and Infectious Diseases (https://www.niaid.nih.gov), NIH-H3Africa (https://h3africa.org) (U01HG007480 and U54HG007480 to C.T.H), the World Bank grant (worldbank.org) (ACE IMPACT project) to C.T.H.The U.S President’s Malaria Initiative (USPMI) funded the primary drug efficacy study from which samples were obtained for the current study

This work is made possible by support from Flu Lab and a cohort of generous donors through TED’s Audacious Project, including the ELMA Foundation, MacKenzie Scott, the Skoll Foundation, and Open Philanthropy. This work was supported by grants from the National Institute of Allergy and Infectious Diseases (https://www.niaid.nih.gov), NIH-H3Africa (https://h3africa.org) (U01HG007480 and U54HG007480 to C.T.H), the World Bank grant (worldbank.org) (ACE IMPACT project) to C.T.H.The U.S President’s Malaria Initiative (USPMI) funded the primary drug efficacy study from which samples were obtained for the current study.

7. We note that Figure 1 in your submission contain [map/satellite] images which may be copyrighted. All PLOS content is published under the Creative Commons Attribution License (CC BY 4.0), which means that the manuscript, images, and Supporting Information files will be freely available online, and any third party is permitted to access, download, copy, distribute, and use these materials in any way, even commercially, with proper attribution. For these reasons, we cannot publish previously copyrighted maps or satellite images created using proprietary data, such as Google software (Google Maps, Street View, and Earth). For more information, see our copyright guidelines: http://journals.plos.org/plosone/s/licenses-and-copyright.

8. Thank you for submitting the above manuscript to PLOS ONE. During our internal evaluation of the manuscript, we found significant text overlap between your submission and the following previously published works.

- https://doi.org/10.1186/s12936-020-03467-3

- https://www.hindawi.com/journals/bmri/2018/2305062/

- https://doi.org/10.1371/journal.pone.0213686

We would like to make you aware that copying extracts from previous publications, especially outside the methods section, word-for-word is unacceptable, even for works which you authored. In addition, the reproduction of text from published reports has implications for the copyright that may apply to the publications.

Please revise the manuscript to rephrase the duplicated text, cite your sources, and provide details as to how the current manuscript advances on previous work. Please note that further consideration is dependent on the submission of a manuscript that addresses these concerns about the overlap in text with published work.

Reviewers' comments:

Reviewer's Responses to Questions

**Comments to the Author**

1. Is the manuscript technically sound, and do the data support the conclusions?

Reviewer #1: Partly

Reviewer #2: Partly

Reviewer #3: Partly

2. Has the statistical analysis been performed appropriately and rigorously? 

Reviewer #1: Yes

Reviewer #2: No

Reviewer #3: Yes

3. Have the authors made all data underlying the findings in their manuscript fully available?

Reviewer #1: Yes

Reviewer #2: Yes

Reviewer #3: Yes

4. Is the manuscript presented in an intelligible fashion and written in standard English?

Reviewer #1: Yes

Reviewer #2: Yes

Reviewer #3: No

5. Review Comments to the Author

Reviewer #1: Artemisinin resistance is a serious threat to malaria control and elimination, and monitoring of antimalarial drug efficacy and resistance is crucial to help policymakers taking informed decisions. The authors present comprehensive data on the prevalence of K13 mutations in different states of Nigeria, and try to understand if the mutations found may have an impact on the efficacy of artemisinin- based combination therapies. However their manuscript has several weaknesses. First the background information is outdated. It is surprising that the authors mention only India when discussing about the spread of that artemisinin resistance outside the Greater Mekong Sub-region. There are reports of validated K13 mutations in South America, but more importantly there a re reports of validated K13 mutations in Rwanda associated with delayed parasite clearance. I think these two publications should be cited : PMID32747827, PMID33864801, and the local emergence of artemisinin resistance in Sub Saharan Africa discussed. The second major limitation of the study is the use of K13 sequences to construct phylogenetic tree. Indeed K13 is under drug pressure and is not the right marker to asses genetic diversity especially in samples collected from a clinical trial. I would recommend the authors to remove the data from the manuscript as they may be misleading. The second comment on the methodology section is the assessment in silico assessment of protein-ligand docking. This manuscript should focus on the monitoring of known and new mutations in K13. The provide information on protein structure is not useful here. There are already many validated or candidate K13 mutations, and the studies looking at the protein-ligand should focus on these mutations. The authors should maybe have used resources to asses molecular markers associated with partner drug resistance, or to establish ex vivo assay to look for new potential markers.

Specific comments

- Introduction, line 58: Artemisinin derivatives were not used in 168 BC, artemisinin was isolated in 1972. In traditional Chinese medicine, Artemisia annua plants were prepared with hot water to treat fever, and this cannot probably lead to resistance

- Introduction, Line 61: The list of validated and candidate mutations in K13 has been continuously updated. please refer to the

last WHO report: https://www.who.int/publications/i/item/9789240012813

- Introduction, lines 63-65: K13 mutations are associated with delayed parasite clearance, not full resistance. When there is no

resistance to partner drug, patients are still cured even with delayed parasite clearance.

- Introduction, lines 65-66: the statement is incorrect. As mentioned in my general comment, delayed parasite clearance was associated with a validated marker (R561H) in Rwanda: PMID33864801

Material and Methods

Could you please give a brief description of the study sites or provide a reference to the original study

Enrollment and sample collection

Could you please mention which drugs were assessed in the original study

Results, demographics

The geometric mean of parasitemia is 17'765 parasites /ul, however in the inclusion criteria, the minimum parasitemia is 20'000 parasites/ul

Table 1

Could you please clarify for which states, the different variables are statistically significant

PfK13 gene mutations and table 2

Please specify which mutations are synonymous and non-synonymous

Table 4

Could you please clarify in table 4 if PPT was recrudescence or new infection.

Discussion

Lines 355-357: The statement is confusing. I guess you mean that the codon 578 is close to 580. But it does not mean that he former mutation is important because it is close to a mutation that is strongly associated with parasite clearance. Several studies have shown that this mutation is not associated with delayed parasite clearance. Whereas its monitoring is important due to its relatively high frequency in sub Saharan Africa, you should not overestimate its importance.

Lines 389-391: You phylogenetic analysis is concluding that you parasites are clustering with parasites from India, Kenya and Uganda. However when looking at table 2, only the widespread A578S was also found in those countries, and not the other mutations found in your studies, except the synonymous mutation (469). How could those mutations be introduced from countries where they did not exist ? Again, this is probably due to the limitation of the marker you used to construct your phylogenetic analysis.

Reviewer #2: This is a molecular survey for alleles in Kelch-13 gene of P. falciparum and not designed to describe the potential emergence and spread of ART resistance. The authors noted that resistance could be via other mechanisms. For a country with over 30 geopolitical regions, the samples is hardly representative of the population of Nigeria. Malaria transmission is overall high in Nigeria but heterogeneous and this could affect the level of drug pressure across the country, allowing for different patterns of evolution of known and emerging drug resistance associated genetic loci. Overall, the study is important as any resistance to artemisinins in Nigeria will seriously hamper hopes of elimination of the disease in Africa.

Major issues

Assent was only for participants at 84-96 months. What of younger children?

Phylogenetic trees are based on mutations rates and not an accurate representation for a recombining organism such as P. falciparum. A neighbour joining tree based on genetic distances would be more appropriate.

BEAST is not optimised for recombination seen in eukaryotic organisms such as P. falciparum and this is a major issue with drawing trees. Several mutation models were determined with the same data and these are not accurate due to the absence of a recombination graph.

What was the rationale for protein ligand docking, as ART does not directly bind to pfk13. This gives the impression of scouting for data rather than a hypothesis orient investigation or a simple survey as applies in this case. The result of docking rather describes the methods. This section is irrelevant and adds nothing to the data.

The highest frequency mutation was less than 2% (not more than 6 individuals overall). In table 4, only C469C was reported in more than 2 individuals. Interestingly this is synonymous variant but seemed to be occurring in individuals with presence of parasitaemia from 3 weeks post commencement of treatment. However, it is hard to consider this number enough to statistically classify the age distribution or correlation with treatment outcome. Being mostly from Enugu, could this be a clonal outbreak carrying the mutation and causing re-infections. There was no indication if these post treatment samples only or were present prior to commencement of treatment

Minor Issues

Introduction

Line 57: The authors stated that ART monotherapy use in China drives drug pressure, but that did not account for the fact that ART-resistant parasite first evolved along the Thai Cambodia border, which is downward on the southern part of the Greater Mekong Sub-region. Perhaps, the escape and spread of ART-resistant isolates from China along the Thai-Cambodia border could have increased the rate of fixation of ART-resistant parasites in that region.

Line 62: Please rephrase, it is the survival rate that is elevated n

Discussion:

The paragraph on protein-ligand docking (starting from line 371) should be combined or commenced with a linking word which contrast the immediate previous paragraph. This is because even though the amino acid substitutions observed have a destabilising effect on the protein structure, they do not necessarily affect drug efficacy.

Reviewer #3: This article by Ajogbasile et al, reports the results of a molecular surveillance of the pfk13 gene conducted in Nigeria in 2018. Previously, some mutations in the propeller part of this gene have been described as correlated with artemisinin resistance in South East Asia, in the Guiana Shield and in some African countries as Rwanda and Ouganda. 332 samples have been analyzed in Nigeria and 13 pfk13 mutations identified in the propeller part of the gene. The manuscript is interested but could be improved if it clearly mentions and associates the TES. The sampling size is also important regarding the previous data available for this country.

However, the manuscript really needs some improvement to better understand the study, the results and to facilitate the reading.

General point to consider:

- The introduction part of the manuscript doesn’t mention the TES study while it is largely described in the method part.

- 300 samples have selected for the study? Two questions: why only 300 out of 586? And why selected them randomly? If you have some responses to treatment and D3 positivity or not, it would be more interesting to select all the interesting profiles regarding the clinical response than entirely randomly selected them.

- Is this TES has been already published? Why not publishing all the results together as they are tightly related?

- The abstract mention line 27 that the paper “evaluated the status of artemisinin resistance”. It is quite ambitious without in vivo or in vitro correlation. In the best case you evaluate the genomic profile of this gene at the time when Act are used in the country.

- The part and figure dedicated to the phylogenic analysis could probably be reduced as less important to answer the question of resistance.

Minor comments:

Introduction line 60 and 346: the information needs to be update. More mutations are validated by WHO.

Line 64 suggests that ARTs are the only effective drug to treat P. falciparum. That is not really true so this assessment should be balanced.

Line 70 mentions that “no systematic molecular epidemiology studies on field …. have so far been conducted”. However, the authors discuss their results according to previous results even if those study had a lower number of samples. Therefore the “no” should be weight.

Figure 1 legend is missing. This figure could be improved if the malaria transmission area is represented and even more, the intensity of transmission.

Table 1: for the genotyping aspect, is this table very useful?

Table 2: it would be easier to read if you order the mutation by position in the gene. To have a complete table, it could also be interesting to mention:

- the total number of mutant samples find for each mutation and alos adding the number of wilt type; In this case, figure 2 is no more useful,

- the TES endpoint at Day 3 and 28 or 42 (mean among the observed cases?),

- if this is the first description of the mutation, if not, are these mutations related with ARTR somewhere else.

Table 3: as the ∆∆G is important it could be useful to classify them from the less stable to the more stable.

No S5 table in the received document.

6. PLOS authors have the option to publish the peer review history of their article (what does this mean?). If published, this will include your full peer review and any attached files.

Reviewer #1: **Yes: **Christian Nsanzabana

Reviewer #2: No

Reviewer #3: No

---

## [Author Response · Author response to Decision Letter 0]

2 Dec 2021

Review Comments to the Author

Reviewer #1

Artemisinin resistance is a serious threat to malaria control and elimination, and monitoring of antimalarial drug efficacy and resistance is crucial to help policymakers make informed decisions. The authors present comprehensive data on the prevalence of K13 mutations in different states of Nigeria, and try to understand if the mutations found may have an impact on the efficacy of artemisinin- based combination therapies. However their manuscript has several weaknesses. First the background information is outdated. It is surprising that the authors mention only India when discussing the spread of that artemisinin resistance outside the Greater Mekong Sub-region. There are reports of validated K13 mutations in South America, but more importantly there are reports of validated K13 mutations in Rwanda associated with delayed parasite clearance. I think these two publications should be cited : PMID32747827, PMID33864801, and the local emergence of artemisinin resistance in Sub Saharan Africa discussed. 

The second major limitation of the study is the use of K13 sequences to construct phylogenetic trees. Indeed K13 is under drug pressure and is not the right marker to assess genetic diversity especially in samples collected from a clinical trial. I would recommend the authors to remove the data from the manuscript as they may be misleading. The second comment on the methodology section is the assessment in silico assessment of protein-ligand docking. This manuscript should focus on the monitoring of known and new mutations in K13. The provide information on protein structure is not useful here. There are already many validated or candidate K13 mutations, and the studies looking at the protein-ligand should focus on these mutations. The authors should maybe have used resources to assess molecular markers associated with partner drug resistance, or to establish ex vivo assay to look for new potential markers.

Response: We appreciate the reviewer for your comments. We have now significantly improved the manuscript and addressed the weaknesses you pointed out as shown in our responses to your specific comments.

Specific comments

- Introduction, line 58: Artemisinin derivatives were not used in 168 BC, artemisinin was isolated in 1972. In traditional Chinese medicine, Artemisia annua plants were prepared with hot water to treat fever, and this cannot probably lead to resistance

 Response: We appreciate the reviewer for this comment. We have now rephrased the introduction. Line 51 - 67 in the revised manuscript now reads …

“Malaria, one of the major global health challenges, has co-existed with humans for over 40 centuries. Control of this ancient disease heavily relies upon the use of antimalarial drugs [1]. Artemisinin-based combination therapies (ACTs) [e.g., artemether-lumefantrine (AL) and artesunate-amodiaquine (AA)] are the current line of treatment for malaria. These drugs were recommended by the World Health Organization (WHO) as first-line treatment for uncomplicated falciparum malaria in 2001 and have since been widely adopted to treat falciparum malaria globally [2]. The Artemisinin (ART)-resistant P. falciparum has been confirmed to have emerged from the Greater Mekong Subregion (GMS) Thai-Cambodian border [3] and has spread to other malaria endemic regions [4 – 7]. 

The pfk13 gene has been reported to be associated with ART-resistance [8, 9]. Several point mutations in the pfk13 gene (F446I, N458Y, M476I, Y493H, R539T, I543T, P553L, R561H, P574L and C580Y) have been validated to correlate with clinical ART resistance in Southeast Asia and South America [2, 10] and confirmed to confer elevated survival rates based on Ring-stage Survival Assays (RSA)0–3 h [11, 12]. Due to these mutations, the efficacy of ACTs may be compromised [13]. Recently for the first time in Africa, two of the validated mutations (R561H and P574L) on the pfk13 gene have been reported in Rwanda to be associated with in vitro resistance to ACTs [6, 7].”

- Introduction, Line 61: The list of validated and candidate mutations in K13 has been continuously updated. please refer to the last WHO report: https://www.who.int/publications/i/item/9789240012813

 Response: We appreciate the reviewer for this comment. This statement has been updated in the manuscript. Line 60 - 63 in the revised manuscript now reads … 

“The pfk13 gene has been reported to be associated with ART-resistance (8, 9). Several point mutations in the pfk13 gene (F446I, N458Y, M476I, Y493H, R539T, I543T, P553L, R561H, P574L and C580Y) have been validated to correlate with clinical ART resistance in Southeast Asia and South America (2,10) …”

- Introduction, lines 63-65: K13 mutations are associated with delayed parasite clearance, not full resistance. When there is no resistance to partner drug, patients are still cured even with delayed parasite clearance.

Response: We appreciate the reviewer for this comment. This statement has now been rephrased in the manuscript. Line 60 - 67 in the revised manuscript now reads …

“The pfk13 gene has been reported to be associated with ART-resistance [8, 9]. Several point mutations in the pfk13 gene (F446I, N458Y, M476I, Y493H, R539T, I543T, P553L, R561H, P574L and C580Y) have been validated to correlate with clinical ART resistance in Southeast Asia and South America [2, 10] and confirmed to confer elevated survival rates based on Ring-stage Survival Assays (RSA)0–3 h [11, 12]. Due to these mutations, the efficacy of ACTs may be compromised [13]. Recently for the first time in Africa, two of the validated mutations (R561H and P574L) on the pfk13 gene have been reported in Rwanda to be associated with in vitro resistance to ACTs [6, 7].”

- Introduction, lines 65-66: the statement is incorrect. As mentioned in my general comment, delayed parasite clearance was associated with a validated marker (R561H) in Rwanda: PMID33864801

Response: We appreciate the reviewer for this comment. As mentioned in our previous response, this statement has now been rephrased.

- Material and Methods

Could you please give a brief description of the study sites or provide a reference to the original study

Response: We appreciate the reviewer for this comment. The link to the original report of the study sites has been provided in the manuscript draft. Line 102 - 103 in the revised manuscript now reads …

“…Full description of study sites is available online at https://www.health.gov.ng/doc/2018-TES-FINAL-REPORT.pdf.”

- Enrollment and sample collection

Could you please mention which drugs were assessed in the original study

Response: We appreciate the reviewer for this comment. However, this information was previously provided in the manuscript. Line 95 - 98 in the manuscript reads …

“This is a retrospective, cross-sectional, community-based study which is part of the therapeutic efficacy study (TES) for monitoring antimalarial efficacies of Artesunate-amodiaquine (AA), Artemether-lumefantrine (AL) and Dihydroartemisinin-piperaquine (DHP) in the treatment of uncomplicated P. falciparum infections in children aged 6-96 months old in Nigeria.” 

- Results, demographics

The geometric mean of parasitemia is 17'765 parasites /ul, however, in the inclusion criteria, the minimum parasitemia is 20'000 parasites/ul

Response: We appreciate the reviewer for this comment. However, the minimum parasitemia stated in the original manuscript draft is 2000 parasites/ul. Line 109 - 113 in the manuscript reads …

“Children were enrolled if they were 6-96 months old. Children were eligible for enrolment if they had symptoms compatible with acute uncomplicated malaria, P. falciparum mono-infection with parasite count ranging from 2,000 - 200,000 asexual forms/μl by microscopy and body (axillary) temperature of ≥ 37.5 °C or a history of fever in the 24 hours preceding presentation.” 

- Table 1

Could you please clarify for which states, the different variables are statistically significant

Response: We appreciate the reviewer for this comment. However, this was previously mentioned in the manuscript. Line 212 - 214 in the manuscript reads ... 

“Children enrolled in Kano State were significantly (p<0.0001) younger and had significantly (p=0.046) lower enrollment asexual parasitaemia compared to children enrolled in other States (Table 1).”

- PfK13 gene mutations and table 2

Please specify which mutations are synonymous and non-synonymous

Response: We appreciate the reviewer for this comment. We have now updated Table 2 in the revised manuscript to indicate synonymous and non-synonymous mutations.

Table 2. Pfk13 polymorphisms observed in Kano, Enugu and Plateau States

S/N K13 Amino Acid Locus Nucleotide Locus Reference Allele Mutant Allele Mutation Type Country Previously Observed Nigerian State Observed

1. K438N 1314 A T Non-synonymous SE Asian Countries Kano

2. G449S 1345 G A Non-synonymous Mali Kano

3. F451L 1353 T - Non-synonymous - Kano

4. C469C 1407 C T Synonymous Kenya, Malawi,

Senegal,

Niger, Congo, DRC Enugu

5. V487E 1460 T A Non-synonymous - Enugu

6. A557S 1669 G T Non-synonymous Congo, DRC, Côte d’Ivoire Enugu

7. A578S 1732 G T Non-synonymous Uganda,

Kenya, DRC, Gabon, Mali, Ghana, Cameroon, Cambodia,

India Enugu

8. Q613H 1839 A T Non-synonymous Senegal, Ghana, Tanzania Enugu & Plateau

9. A621A 1863 T A Synonymous - Enugu

10. Q661H 1983 A - Non-synonymous - Enugu

11. N664I 1991 A T Non-synonymous - Plateau

12. V692G 2075 T G Non-synonymous - Enugu & Plateau

13. N694K 2082 T A Non-synonymous Angola, Cote d’Ivoire Kano

- Table 4

Could you please clarify in table 4 if PPT was recrudescence or new infection.

Response: We appreciate the reviewer for this comment. Table 4 has now been updated in the revised manuscript. 

Table 4: Clinical features of children with mutated P falciparum in this study.

Study site Sample ID Gender Age (Month) Enrollment asexual parasitaemia (uL-1) Mutation Position Parasite clearance time (day) Antimalarial Treatment Treatment outcome

Enugu 33 M 48 30571 A578S 1 AL ACPR

 89 F 60 12817 Q613H 2 AL ACPR

 106 F 36 16290 Q613H 2 AL ACPR

 134 M 72 57850 A621A 2 AL ACPR

 154 M 60 3424 V487E 2 AL ACPR

 179 M 36 61320 A557S 2 DHP ACPR

 197 M 48 2055 C469C 2 DHP PPT

(D42)

 228+

 F 24 91680 C469C 2 AL PPT

(D21)

 236 M 36 9794 C469C 2 AL PPT

(D28)

 239 M 72 6714 V692G

Q661H 1 AL PPT

(D28)

 254 M 96 29229 C469C 2 AL PPT

(D28)

Kano 

 14 M 12 15923 N694K 2 AA ACPR

 26 F 24 37880 F451L 1 AL ACPR

 51 F 84 26523 K438N 2 AA ACPR

 59 M 36 16888 G449S 2 AL ACPR

 77 M 48 2250 G449S 1 AA ACPR

Plateau 31 F 84 26069 Q613H 1 AA ACPR

 58 M 60 16304 Q613H 1 AL ACPR

 62 F 72 61080 Q613H 2 AL ACPR

 77 F 60 5428 N664I 1 AL PPT

 (D28)

 155 M 60 54739 V692G 2 AA PPT

(D28)

Mean (SD) 53.71

(22) 17, 530* 1.67 (0.48) 29

(6.3)#

*Geometric mean; #Mean PPT: Parasite post treatment, AA: Artesunate-amodiaquine; AL: Artemether-lumefantrine; DHP: Dihydroartemisinin-piperaquine, ACPR: Adequate clinical and parasitological response

+Only sample with recrudescence parasitaemia

- Discussion

Lines 355-357: The statement is confusing. I guess you mean that the codon 578 is close to 580. But it does not mean that the former mutation is important because it is close to a mutation that is strongly associated with parasite clearance. Several studies have shown that this mutation is not associated with delayed parasite clearance. Whereas its monitoring is important due to its relatively high frequency in sub Saharan Africa, you should not overestimate its importance.

Response: We appreciate the reviewer for this comment. We have now adjusted this statement in the revised version of the manuscript. Line 306 - 315 in the revised manuscript now reads … 

“In addition, eight (A578S, C469C, Q613H, K438N, A621A, N694K, G499S and A557S) of the 13 SNPs observed in our study have also been observed in Southeast Asia and other sub-Saharan African countries [8, 20, 21, 24, 26, 27, 29, 32, 33, 37 – 41] as represented in Table 3. The Q613H was the most prominent SNP observed in our study and was also reported in recent studies in Nigeria [22, 23]. A578S is, however the most prominent SNP observed among sub-Saharan African countries such as Ghana, Kenya, Gabon, DRC, Uganda, Cameroon and Mali [27, 32, 40]. This mutation (A578S) represents a change from a non-polar to a polar amino acid and can alter the shape of the pfk13 protein in the regions it has been observed in [28]. Therefore, more attention must be paid to this mutation as it is emerging to be the most prominent mutation observed in the pfk13 gene in sub-Saharan Africa [27, 32, 40].”

- Lines 389-391: Your phylogenetic analysis is concluding that you parasites are clustering with parasites from India, Kenya and Uganda. However when looking at table 2, only the widespread A578S was also found in those countries, and not the other mutations found in your studies, except the synonymous mutation (469). How could those mutations be introduced from countries where they did not exist ? Again, this is probably due to the limitation of the marker you used to construct your phylogenetic analysis.

Response: We appreciate the reviewer for this comment. We have now removed the phylogenetic analysis from the manuscript.

Reviewer #2

This is a molecular survey for alleles in Kelch-13 gene of P. falciparum and not designed to describe the potential emergence and spread of ART resistance. The authors noted that resistance could be via other mechanisms. For a country with over 30 geopolitical regions, the sample is hardly representative of the population of Nigeria. Malaria transmission is overall high in Nigeria but heterogeneous and this could affect the level of drug pressure across the country, allowing for different patterns of evolution of known and emerging drug resistance associated with genetic loci. Overall, the study is important as any resistance to artemisinins in Nigeria will seriously hamper hopes of elimination of the disease in Africa.

Major issues

- Assent was only for participants at 84-96 months. What of younger children?

Response: We appreciate the reviewer for this comment. However, for such children, informed consent was obtained from parents or legal guardians. 

- Phylogenetic trees are based on mutation rates and not an accurate representation for a recombinant organism such as P. falciparum. A neighbour-joining tree based on genetic distances would be more appropriate.

Response: We appreciate the reviewer for this comment. We have removed the phylogenetic analyses from the manuscript based on your recommendation and that of the other reviewers. 

- BEAST is not optimised for recombination seen in eukaryotic organisms such as P. falciparum and this is a major issue with drawing trees. Several mutation models were determined with the same data and these are not accurate due to the absence of a recombination graph.

Response: We appreciate the reviewer for this comment. We have removed BEAST analysis from the manuscript based on your recommendation and that of the other reviewers. 

- What was the rationale for protein ligand docking, as ART does not directly bind to pfk13. This gives the impression of scouting for data rather than a hypothesis orient investigation or a simple survey as applied in this case. The result of docking rather describes the methods. This section is irrelevant and adds nothing to the data.

Response: We appreciate the reviewer for this comment. We have now removed the protein-ligand docking analysis from the manuscript.

- The highest frequency mutation was less than 2% (not more than 6 individuals overall). In table 4, only C469C was reported in more than 2 individuals. Interestingly this is a synonymous variant but seemed to be occurring in individuals with presence of parasitaemia from 3 weeks post commencement of treatment. However, it is hard to consider this number enough to statistically classify the age distribution or correlation with treatment outcome. Being mostly from Enugu, could this be a clonal outbreak carrying the mutation and causing re-infections. There was no indication if these post treatment samples only or were present prior to commencement of treatment

Response: Thank you for your thoughts on the clonal outbreak which could be the case however, we did not conduct a high resolution experiment to investigate this possibility. Also, the C469C SNP was only present in post treatment samples from Enugu.

Minor Issues

- Introduction

Line 57: The authors stated that ART monotherapy use in China drives drug pressure, but that did not account for the fact that ART-resistant parasites first evolved along the Thai Cambodia border, which is downward on the southern part of the Greater Mekong Sub-region. Perhaps, the escape and spread of ART-resistant isolates from China along the Thai-Cambodia border could have increased the rate of fixation of ART-resistant parasites in that region.

Response: We appreciate the reviewer for this comment. We have now rephrased the introduction.

- Line 62: Please rephrase, it is the survival rate that is elevated n

Response: We appreciate the reviewer for this comment. We have rephrased this statement in the manuscript. Line 60 - 64 in the revised manuscript now reads …

“The pfk13 gene has been reported to be associated with ART-resistance [8, 9]. Several point mutations in the pfk13 gene (F446I, N458Y, M476I, Y493H, R539T, I543T, P553L, R561H, P574L and C580Y) have been validated to correlate with clinical ART resistance in Southeast Asia and South America [2, 10] and confirmed to confer elevated survival rates based on Ring-stage Survival Assays (RSA)0–3 h [11, 12].”

- Discussion:

The paragraph on protein-ligand docking (starting from line 371) should be combined or commenced with a linking word which contrasts the immediate previous paragraph. This is because even though the amino acid substitutions observed have a destabilising effect on the protein structure, they do not necessarily affect drug efficacy.

Response: We appreciate the reviewer for this comment. We have now removed the protein-ligand docking analysis and corresponding discussions from the manuscript. 

Reviewer #3

This article by Ajogbasile et al, reports the results of a molecular surveillance of the pfk13 gene conducted in Nigeria in 2018. Previously, some mutations in the propeller part of this gene have been described as correlated with artemisinin resistance in South East Asia, in the Guiana Shield and in some African countries as Rwanda and Uganda. 332 samples have been analyzed in Nigeria and 13 pfk13 mutations identified in the propeller part of the gene. The manuscript is interesting but could be improved if it clearly mentions and associates the TES. The sampling size is also important regarding the previous data available for this country.

However, the manuscript really needs some improvement to better understand the study, the results and to facilitate the reading.

General point to consider:

- The introduction part of the manuscript doesn’t mention the TES study while it is largely described in the method part.

Response: We appreciate the reviewer for this comment. We have now made significant changes to the introduction in the revised manuscript and made mention of the TES study.

- 300 samples have been selected for the study? Two questions: why only 300 out of 586? 

Response: We only had funding for 300 samples. As a result, we selected 300 samples for molecular analysis. 

- And why selected them randomly? If you have some responses to treatment and D3 positivity or not, it would be more interesting to select all the interesting profiles regarding the clinical response than entirely randomly selecting them.

Response: Of the 586 patients enrolled, 32 patients had treatment failure across the three states and these samples were selected. After which a random selection of samples across the three states was carried out to make up to 100 samples per state. 

- Is this TES has been already published? Why not publish all the results together as they are tightly related?

Response: Yes, the TES study has been published. Our current study was centered on molecular surveillance for polymorphisms associated with ART. The TES that was published focused on the clinical responses of patients to the drugs used. 

- The abstract mentions in line 27 that the paper “evaluated the status of artemisinin resistance”. It is quite ambitious without in vivo or in vitro correlation. In the best case you evaluate the genomic profile of this gene at the time when Act is used in the country.

Response: We appreciate the reviewer for this comment. We have rephrased this statement in the manuscript. Line 29 - 31 in the revised manuscript now reads

“In this study, we evaluated the genomic profile of the pfk13 gene associated with artemisinin resistance in Plasmodium falciparum in Nigerian children, by targeted sequencing of the pfk13 gene.”

- The part and figure dedicated to the phylogenetic analysis could probably be reduced as less important to answer the question of resistance.

Response: We appreciate the reviewer for this comment. We have now removed the section on phylogenetic analysis based on the recommendations of the other two reviewers. 

Minor comments:

- Introduction line 60 and 346: the information needs to be updated. More mutations are validated by WHO.

Response: We appreciate this reviewer for this important observation. The introduction has been updated to include these validated mutations. Line 60 - 63 in the revised manuscript now reads … 

“The pfk13 gene has been reported to be associated with ART-resistance [8, 9]. Several point mutations in the pfk13 gene (F446I, N458Y, M476I, Y493H, R539T, I543T, P553L, R561H, P574L and C580Y) have been validated to correlate with clinical ART resistance in Southeast Asia and South America [2,10].”

- Line 64 suggests that ARTs are the only effective drug to treat P. falciparum. That is not really true so this assessment should be balanced.

Response: We appreciate the reviewer for this comment. We have rephrased the statement in the manuscript. Line 64 - 65 in the revised manuscript now reads …

 “Due to these mutations, the efficacy of ACTs may be compromised (13).”

- Line 70 mentions that “no systematic molecular epidemiology studies on field …. have so far been conducted”. However, the authors discuss their results according to previous results even if those studies had a lower number of samples. Therefore the “no” should be weight.

Response: We appreciate the reviewer for this comment. We have rephrased the statement in the manuscript. Line 70 - 72 in the revised manuscript now reads … 

“Despite the need to regularly survey for emergence of these pfk13 mutant alleles in different malaria-endemic regions of Nigeria, only a small number of systematic molecular epidemiological studies on field P. falciparum isolates have so far been conducted [2].”

- Figure 1 legend is missing. This figure could be improved if the malaria transmission area is represented and even more, the intensity of transmission.

Response: We appreciate the reviewer for this comment. We have now removed the previous figure 1 from the manuscript due to copyright issues raised by the journal.

- Table 1: for the genotyping aspect, is this table very useful?

Response: We appreciate the reviewer for this comment. However, the purpose of Table 1 is to give readers a brief description of the sample demographics for this study. 

- Table 2: it would be easier to read if you order the mutation by position in the gene. To have a complete table, it could also be interesting to mention: - the total number of mutant samples find for each mutation and also adding the number of wild type; In this case, figure 2 is no more useful,

Response: We appreciate the reviewer for this comment. We have now ordered mutations by position in the gene as suggested. However, we decided to keep figure 2 (now figure 1 in the revised manuscript) because we feel it gives a more interactive view of the percentage of mutations and wild type. 

- the TES endpoint at Day 3 and 28 or 42 (mean among the observed cases?)

Response: This question is unclear to us. However, if the reviewer is asking if values from these days are means from all samples that had these days of failure, then no they are not. 

- if this is the first description of the mutation, if not, are these mutations related with ART R somewhere else.

Response: We have reported some novel mutations that have not been reported elsewhere. As such we cannot say if these mutations are associated with ART resistance. Further molecular and in vivo assays need to be done to validate the role of these mutations in ART resistance.

- Table 3: as the ∆∆G is important it could be useful to classify them from the less stable to the more stable.

Response: We appreciate the reviewer for this comment. We have now removed the protein stability analysis as suggested by the other two reviewers.

- No S5 table in the received document.

Response: This was submitted in the original document. This was a table showing K13 mutations in Nigeria (including the ones observed in this study). However, this is now Table 3 in the revised manuscript.

Table 3: Molecular Surveillance of Pfk13 propeller polymorphisms in Nigeria

S/N Mutations Previously observed Year observed Reference

1 G496G 2014 [20]

2 K610R 2014 [20]

3 A626T 2014 [20]

4 A627A 2014 [20]

5 P441S 2017, 2018 [21]

6 C469C* 2017 [21]

7 V589V 2016 [21]

8 V650F 2016 [21]

9 N664N 2017 [21]

10 A676A 2016 [21]

11 Q613H* 2010, 2011, 2015, 2016 [22, 23]

12 K189T 2015, 2016 [23]

13 H136N 2015, 2016 [23]

14 D464N 2010, 2011 [22]

15 G665C 2016 [13]

16 V666V 2016 [13]

17 P553P 2016 [13]

18 V510V 2016 [13]

19 A578S* 2010, 2011 [22]

20 A621A* 2014 [24]

21 K438N 2018 Observed in this study

22 G449S 2018 Observed in this study

23 F451L 2018 Observed in this study

24 N694K 2018 Observed in this study

25 A557S 2018 Observed in this study

26 V487E 2018 Observed in this study

27 V692G 2018 Observed in this study

28 Q661H 2018 Observed in this study

29 N664I 2018 Observed in this study

*Mutations observed in previous studies and our study

---

## [Editor Report · Decision Letter 1]

17 Jan 2022

PONE-D-21-25383R1Molecular profiling of artemisinin resistance Kelch 13 gene in Plasmodium falciparum from NigeriaPLOS ONE

We look forward to receiving your revised manuscript.

Kind regards,

Jutta Marfurt, PhD

Academic Editor

PLOS ONE

Journal Requirements:

Additional Editor Comments (if provided):

Comments/suggested changes from the editor:

General:

Please use abbreviations consistently throughout the manuscript and write them in full the first time they appear in the manuscript. There are also a couple of typos and some grammar errors which should be corrected.

Title:

1. Change to “Molecular profiling of the artemisinin resistance Kelch 13 gene in Plasmodium falciparum from Nigeria”

Introduction:

2. Line 57: Delete “The” before “Artemisinin-resistant”.

3. Line 59: Also cite doi: 10.1056/NEJMc0805011 alongside REF #3

4. Line 60: The first sentence should be deleted.

5. Line 64: Cite REF #8 alongside REFs #11 and #12.

6. Lines 64-65: “Due to these mutations, the efficacy of ACTs may be compromised”. REF #13 is inappropriate. Furthermore, I would expect some lines about partner drug resistance here. See also comment of Reviewer #1.

7. Line 69: Delete REF #15 (Colombian study).

Materials and Methods:

8. Line 141: iii) Asexual parasite clearance time (PCT) to line 142. I also suggest including the commonly used terms ACPR, ETF, LCF, and LPF, respectively, as used in the final TES report. These would not have to be described/defined in detail in the manuscript; the official WHO TES study protocol can be cited instead.

9. Lines 145-146: This sentence needs to be reworded.

Results:

10. Between “Demographics” and Pfk13 gene mutations”, a short summary of the TES results (i.e., summary table of crude and PCR-corrected ACPR, ETF, LCF, and LPF) of the 300 samples would be very helpful.

11. Lines 222-224: Replace with: “Thirteen pfk13 gene mutations were detected in 21 out of the 332 sequences analyzed in this study (Table 2).”

12. Line 225: Delete “and”.

13. In Figure 1 and Table 3, please order the SNPs in ascending order according to the location in the gene.

14. Lines 226 and 227: Replace “sequences” with “samples”.

15. Table 3 was referred to in line 224. However, this is not correct because the text outlined the results of the current study. Table 3 summarises the results of all Nigerian studies hitherto conducted; this should be mentioned in the text. On another note: Is there a specific reason why the polymorphisms reported by Abubakar et al. (2020; DOI: 10.3390/tropicalmed5020085) are not included in Table 3?

16. Line 251: “…(mean time to recurrence: 29±6.3 days).”

17. Footnote Table 4/Line 263: “…recurrent parasitaemia”

18. Lines 266-267: Replace with “Comparison of responsiveness indices following treatment initiation in children with and without mutated Pfk13.”

Discussion:

19. The first sentence of the Discussion should be toned down, particularly in view of the fact that no SNPs associated with artemisinin resistance were observed and nothing is reported on partner drug resistance.

20. Line 300: Replace “is” with “was”.

21. Line 317: Replace “has” with “have”.

22. Line 322: “…are less likely to be associated with a delayed parasite clearance phenotype…”

23. Lines 324-326: “In addition, significantly longer asexual parasite clearance times in children infected with non-mutant Pfk13 parasites indicate that mutants identified in the parasites circulating in Nigeria do not confer resistance to artemisinin derivatives.

Supporting Information:

24. The Supporting Information could be omitted. Instead, a reference to the WHO protocol for parasite genotyping to differentiate recrudescence from new infections could be in included in the footnote of the summary table I suggested in comment # 10.
---

## [Author Response · Author response to Decision Letter 1]

26 Jan 2022

Comments/suggested changes from the editor:

General:

Please use abbreviations consistently throughout the manuscript and write them in full the first time they appear in the manuscript. There are also a couple of typos and some grammar errors which should be corrected.

Response: We appreciate the editor for your thorough review of our manuscript. We have taken note of your comments and made the necessary corrections. 

Title:

1. Change to “Molecular profiling of the artemisinin resistance Kelch 13 gene in Plasmodium falciparum from Nigeria”

Response: We appreciate the editor for your comment. We have now changed the title of the manuscript to “Molecular profiling of the artemisinin resistance Kelch 13 gene in Plasmodium falciparum from Nigeria” as suggested. Line 1 - 2 in the revised manuscript now reads …

“Full Title: Molecular profiling of the artemisinin resistance Kelch 13 gene in Plasmodium falciparum from Nigeria”

 Introduction: 

2. Line 57: Delete “The” before “Artemisinin-resistant”.

Response: We have removed ‘The’ from the statement. Line 57 in the revised manuscript now reads …

“...Artemisinin (ART)-resistant P. falciparum has been…”

3. Line 59: Also cite doi: 10.1056/NEJMc0805011 alongside REF #3

Response: This has now been cited in the manuscript. Line 59 in the revised manuscript now reads …

“border [3, 4] and has spread to other malaria endemic regions [5 – 8]. 

4. Line 60: The first sentence should be deleted.

Response: This sentence has been deleted. Line 60 in the revised manuscript now reads …

“Several point mutations in the pfk13 gene (F446I, N458Y, M476I, Y493H, R539T, I543T,”

5. Line 64: Cite REF #8 alongside REFs #11 and #12.

Response: We thank the editor for your comment. Line 63 in the revised manuscript now reads … 

“...survival rates based on Ring-stage Survival Assays (RSA)0–3 h [10-12].” 

6. Lines 64-65: “Due to these mutations, the efficacy of ACTs may be compromised”. REF #13 is inappropriate. Furthermore, I would expect some lines about partner drug resistance here. See also comment of Reviewer #1.

Response: We appreciate the editor for your comment. We have now changed the reference in the revised manuscript and also added a few statements on partner drug resistance. Line 63 - 68 in the revised manuscript now reads …

“Furthermore, mutations and increased copy numbers of genes such as P. falciparum multidrug resistant gene-1 (pfmdr1) and P. falciparum chloroquine resistance transporter (pfcrt) gene have been linked to resistance in artemisinin (and derivatives) partner drugs such as lumefantrine, amodiaquine, mefloquine and piperaquine [13-18]. Due to these mutations, the efficacy of ACTs may be compromised [19].”

7. Line 69: Delete REF #15 (Colombian study).

Response: This has been deleted. Line 73 in the revised manuscript now reads …

“no association between ART-resistant parasites and the ten validated mutations has been found [20-23].” 

Materials and Methods:

8. Line 141: iii) Asexual parasite clearance time (PCT) to line 142. I also suggest including the commonly used terms ACPR, ETF, LCF, and LPF, respectively, as used in the final TES report. These would not have to be described/defined in detail in the manuscript; the official WHO TES study protocol can be cited instead.

Response: We appreciate the editor for your comment. We have now included the ACPR, ETF, LCF and LPF as you suggested. Line 142 - 155 in the revised manuscript now reads …

“Response to drug treatment was evaluated using the following treatment indices: 

(i) Adequate clinical and parasitological response (ACPR), early treatment failure (ETF), late clinical failure (LCF) and late parasitological failure (LPF) [Full description of WHO TES study protocol available at https://www.who.int/docs/default-source/documents/publications/gmp/methods-for-surveillance-of-antimalarial-drug-efficacy.pdf?sfvrsn=29076702_2].

(ii) Asexual parasite reduction ratio (PRR) one or two days after treatment initiation (PRRD1 or PRRD2) defined as the ratio of asexual parasitaemia pre-treatment initiation and that on day one or two, respectively.

(iii) Asexual parasite positivity on day 1 (APPD1), 2 (APPD2) or 3 (APPD3) defined as proportion of children with residual asexual parasitaemia one, two or three days after treatment initiation, respectively Asexual parasite clearance time (PCT) defined as the time elapsing between drug administration and absence of microscopic detection of viable asexual parasitaemia.”

9. Lines 145-146: This sentence needs to be reworded.

Response: We have now reworded this statement in the manuscript. Line 159 - 161 in the revised manuscript now reads …

“A total of 300 pre-treatment (Day 0) DBS filter paper samples from all three sites were utilised in this study. Of the 300 samples, 32 had LPF making a final total of 332 samples selected for analysis.”

Results:

10. Between “Demographics” and Pfk13 gene mutations”, a short summary of the TES results (i.e., summary table of crude and PCR-corrected ACPR, ETF, LCF, and LPF) of the 300 samples would be very helpful.

Response: We appreciate the editor for your comment. We have now included a table describing the summary of the TES study as Table 2. Table 2 in the revised manuscript now reads …

“Table 2: Summary of treatment outcome by State and drug

State Treatment

Outcome Drug

AA AL DHP Total

Enugu Number of samples

ETF

LCF

LPF

ACPR_u

ACPR_c

%ACPR_c 50 50

 0 0

 0 0

 11 2

 39 48

 49 50

 98 100 100

0

0

14

87

99

99

Kano Number of samples

ETF

LCF

LPF

ACPR_u

ACPR_c

%ACPR_c 50 50

 0 0 

 0 0

 2 9 

48 41 

50 50

100 100 100

0

0

11

89

100

100

Plateau Number of samples

ETF

LCF

LPF

ACPR_u

ACPR_c

%ACPR_c 50 50

 0 0 

 0 0 

 1 7

49 43 

50 50

100 100 100

0

0

8

92

100

100

Total Number of samples

ETF

LCF

LPF

ACPR_u

ACPR_c

%ACPR_c 100 150 50

 0 0 0 

 0 0 0 

 3 27 2

97 123 47 

100 149 50

100 99.3 100 300

0

0

32

267

299

99.7

Crude ACPR (ACPR_u); PCR-corrected ACPR (ACPR_c)

WHO protocol for parasite genotyping to differentiate recrudescence from new infections available at http://whqlibdoc.who.int/publications/2008/9789241596305_eng.pdf

11. Lines 222-224: Replace with: “Thirteen pfk13 gene mutations were detected in 21 out of the 332 sequences analyzed in this study (Table 2).”

Response: We have effected this change in the manuscript. Line 240 - 241 in the revised manuscript now reads …

“...Thirteen pfk13 gene mutations were detected in 21 out of the 332 sequences analysed in this study (Table 3)...” 

12. Line 225: Delete “and”.

Response: This has been deleted. Line 242 in the revised manuscript now reads … 

“chromosome 13 in the propeller region of the pfk13 protein was 6.3%, 93.7% did not have…”

13. In Figure 1 and Table 3, please order the SNPs in ascending order according to the location in the gene.

Response: We appreciate the editor for your comment. We have now ordered the SNPs in ascending order according to the location in the gene in Figure 1 and Table 3 (now Table 4 in the revised manuscript). Figure 1 and Table 4 in the revised manuscript now reads thus …

Fig 1. A chart showing the frequency distribution of pfk13 gene polymorphisms observed in this study

Table 4: Molecular Surveillance of Pfk13 propeller polymorphisms in Nigeria

S/N Mutations Previously observed Year observed Reference

1 H136N 2015, 2016 [26]

2 K189T 2015, 2016 [26]

3 E433G 2018 [27]

4 F434I 2018 [27]

5 F434S 2018 [27]

6 K438N 2018 Observed in this study

7 P441S 2017, 2018 [28]

8 F442F 2018 [27]

9 G449S 2018 Observed in this study

10 F451L 2018 Observed in this study

11 D464N 2010, 2011 [29]

12 C469C* 2017 [28]

13 V487E 2018 Observed in this study

14 F492F 2018 [27]

15 G496G 2014 [30]

16 V510V 2016 [20]

17 P553P 2016 [20]

18 A557S 2018 Observed in this study

19 A578S* 2010, 2011 [29]

20 V589V 2016 [28]

21 K610R 2014 [30]

22 Q613H* 2010, 2011, 2015, 2016 [29, 26]

23 A621A* 2014 [31]

24 A626T 2014 [30]

25 A627A 2014 [30]

26 V650F 2016 [28]

27 Q661H 2018 Observed in this study

28 N664I 2018 Observed in this study

29 N664N 2017 [28]

30 G665C 2016 [20]

31 V666V 2016 [20]

32 A676A 2016 [28]

33 I684N 2018 [27]

34 I684T 2018 [27]

35 E688K 2018 [27]

36 V692G 2018 Observed in this study

37 N694K 2018 Observed in this study

*Mutations observed in previous studies and our study

14. Lines 226 and 227: Replace “sequences” with “samples”.

Response: This change has been effected in the manuscript. Line 243 - 245 in the revised manuscript now reads …

“The highest occurring mutation Q613H (1.5%) was detected in five samples (three from Plateau State and two from Enugu State). The C469C mutation with 1.2% was detected in four samples from Enugu State.”

15. Table 3 was referred to in line 224. However, this is not correct because the text outlined the results of the current study. Table 3 summarises the results of all Nigerian studies hitherto conducted; this should be mentioned in the text. On another note: Is there a specific reason why the polymorphisms reported by Abubakar et al. (2020; DOI:10.3390/tropicalmed5020085) are not included in Table 3?

Response: We appreciate the editor for this comment. We have now correctly referred to Table 3 (now Table 4 in the revised manuscript) in the manuscript. There was no particular reason why we didn’t report the polymorphisms reported by Abubakar et al. , it was an omission on our part. However, we have now included the polymorphisms in our table 4. Line 246 - 247 in the revised manuscript now reads …

“…We summarized mutations detected in this study and previous Nigerian studies in Table 4.”

Table 4: Molecular Surveillance of Pfk13 propeller polymorphisms in Nigeria

S/N Mutations Previously observed Year observed Reference

1 H136N 2015, 2016 [26]

2 K189T 2015, 2016 [26]

3 E433G 2018 [27]

4 F434I 2018 [27]

5 F434S 2018 [27]

6 K438N 2018 Observed in this study

7 P441S 2017, 2018 [28]

8 F442F 2018 [27]

9 G449S 2018 Observed in this study

10 F451L 2018 Observed in this study

11 D464N 2010, 2011 [29]

12 C469C* 2017 [28]

13 V487E 2018 Observed in this study

14 F492F 2018 [27]

15 G496G 2014 [30]

16 V510V 2016 [20]

17 P553P 2016 [20]

18 A557S 2018 Observed in this study

19 A578S* 2010, 2011 [29]

20 V589V 2016 [28]

21 K610R 2014 [30]

22 Q613H* 2010, 2011, 2015, 2016 [29, 26]

23 A621A* 2014 [31]

24 A626T 2014 [30]

25 A627A 2014 [30]

26 V650F 2016 [28]

27 Q661H 2018 Observed in this study

28 N664I 2018 Observed in this study

29 N664N 2017 [28]

30 G665C 2016 [20]

31 V666V 2016 [20]

32 A676A 2016 [28]

33 I684N 2018 [27]

34 I684T 2018 [27]

35 E688K 2018 [27]

36 V692G 2018 Observed in this study

37 N694K 2018 Observed in this study

*Mutations observed in previous studies and our study

16. Line 251: “...(mean time to recurrence: 29±6.3 days).”

Response: This has been corrected in the manuscript. Line 279 - 280 in the revised manuscript now reads …

“within 21-42 days of follow-up post treatment initiation (mean time to recurrence: 29±6.3 days),…”

17. Footnote Table 4/Line 263: “...recurrent parasitaemia”

Response: This has been corrected in the manuscript. Line 286 in the revised manuscript now reads …

“ *Geometric mean; #Mean LPF;+Only sample with recrudescence recurrent parasitaemia”

18. Lines 266-267: Replace with “Comparison of responsiveness indices following treatment

initiation in children with and without mutated Pfk13.”

Response: We appreciate the editor for your comment. This has now been corrected in the manuscript. Line 290 - 291 in the revised manuscript now reads …

“Comparison of responsiveness indices following treatment initiation in children with and without mutated Pfk13”

Discussion:

19. The first sentence of the Discussion should be toned down, particularly in view of the fact that no SNPs associated with artemisinin resistance were observed and nothing is reported on partner drug resistance.

Response: We appreciate the editor for this comment. We have now rephrased this statement in the manuscript. Line 303 - 305 in the revised manuscript now reads ..

“The knowledge of mutations in the pfk13 gene associated with slow clearance of artemisinin derivatives provides the ability to track the emergence and prevent the spread of resistant parasites and assess the effectiveness of control measures.”

20. Line 300: Replace “is” with “was”.

Response: This has been changed in the manuscript. Line 307 in the revised manuscript now reads …

“...None of them was among the ten” 

21. Line 317: Replace “has” with “have”.

Response: This has been changed in the manuscript. Line 323 in the revised manuscript now reads …

“of our knowledge, have not yet been described elsewhere…”

22. Line 322: “...are less likely to be associated with a delayed parasite clearance phenotype...”

Response: This has been changed in the manuscript. Line 328 in the revised manuscript now reads …

“are less likely to be associated with a delayed parasite clearance phenotype as evidenced by” 

23. Lines 324-326: “In addition, significantly longer asexual parasite clearance times in children infected with non-mutant Pfk13 parasites indicate that mutants identified in the parasites circulating in Nigeria do not confer resistance to artemisinin derivatives.

Response: This has been changed in the manuscript. Line 330 - 332 in the revised manuscript now reads …

“In addition, significantly longer asexual PCT in children infected with non-mutant pfk13 parasites indicate the mutants identified in the parasites circulating in Nigeria do not confer resistance to artemisinin derivatives.”

Supporting Information:

24. The Supporting Information could be omitted. Instead, a reference to the WHO protocol for parasite genotyping to differentiate recrudescence from new infections could be in included in the footnote of the summary table I suggested in comment # 10. 

Response: We appreciate the editor for your comment. The supporting information has now been removed and a reference to the WHO protocol has been included as a footnote to Table 2 as suggested. Table 2 in the revised manuscript now reads thus …

Table 2: Summary of treatment outcome by State and drug

State Treatment

Outcome Drug

AA AL DHP Total

Enugu Number of samples

ETF

LCF

LPF

ACPR_u

ACPR_c

%ACPR_c 50 50

 0 0

 0 0

 11 2

 39 48

 49 50

 98 100 100

0

0

14

87

99

99

Kano Number of samples

ETF

LCF

LPF

ACPR_u

ACPR_c

%ACPR_c 50 50

 0 0 

 0 0

 2 9 

48 41 

50 50

100 100 100

0

0

11

89

100

100

Plateau Number of samples

ETF

LCF

LPF

ACPR_u

ACPR_c

%ACPR_c 50 50

 0 0 

 0 0 

 1 7

49 43 

50 50

100 100 100

0

0

8

92

100

100

Total Number of samples

ETF

LCF

LPF

ACPR_u

ACPR_c

%ACPR_c 100 150 50

 0 0 0 

 0 0 0 

 3 27 2

97 123 47 

100 149 50

100 99.3 100 300

0

0

32

267

299

99.7

Crude ACPR (ACPR_u); PCR-corrected ACPR (ACPR_c)

WHO protocol for parasite genotyping to differentiate recrudescence from new infections available at http://whqlibdoc.who.int/publications/2008/9789241596305_eng.pdf

---

## [Editor Report · Decision Letter 2]

14 Feb 2022

Molecular profiling of the artemisinin resistance Kelch 13 gene in Plasmodium falciparum from Nigeria

PONE-D-21-25383R2

Dear Dr. Happi,

We’re pleased to inform you that your manuscript has been judged scientifically suitable for publication and will be formally accepted for publication once it meets all outstanding technical requirements.

Kind regards,

Jutta Marfurt, PhD

Academic Editor

PLOS ONE
---

## [Editor Report · Acceptance letter]

18 Feb 2022

PONE-D-21-25383R2 

Molecular profiling of the artemisinin resistance Kelch 13 gene in *Plasmodium falciparum* from Nigeria 

Dear Dr. Happi:

I'm pleased to inform you that your manuscript has been deemed suitable for publication in PLOS ONE. Congratulations! Your manuscript is now with our production department. 

Kind regards, 

on behalf of

Dr. Jutta Marfurt 

Academic Editor

PLOS ONE